# Exploring barriers and enablers to diabetes self-care practice in Ethiopia, 2025: A qualitative systematic review

Sadik Abdulwehab[1]*, Frezer Kedir[2]

**1** School of Nursing, Wollega University, Nekemte, Oromia, Ethiopia, **2** School of Nursing, Jimma University, Jimma, Southwest Oromia, Ethiopia

* sadikabdulwehab@gmail.com

## Abstract

### Introduction

Diabetes mellitus is a growing public health concern in Ethiopia, with increasing prevalence and a high proportion of undiagnosed cases. Effective self-care is crucial for managing diabetes; however, many patients face challenges ranging from personal beliefs to systemic and sociocultural constraints. Although multiple qualitative studies have explored these challenges, a synthesized, context-specific understanding that incorporates broader social and health system factors remains limited. This review aimed to synthesize qualitative evidence on barriers and facilitators of diabetes self-care practice in Ethiopia.

### Methods

A qualitative systematic review was conducted in accordance with PRISMA 2020 and PRISMA-QS guidelines. A comprehensive search of PubMed/MEDLINE, Scopus, CINAHL, Web of Science, and Google Scholar was performed using relevant keywords and Medical Subject Headings. Eligible studies included primary qualitative and mixed-methods studies with extractable qualitative findings that explored diabetes self-care barriers and facilitators among patients, caregivers, or healthcare providers in Ethiopia. Purely quantitative studies, reviews, editorials, and studies conducted outside Ethiopia were excluded. Thematic synthesis was employed to integrate findings, and the GRADE-CERQual approach was used to assess confidence in the review findings.

### Results

Eleven studies were included, comprising qualitative and mixed-methods designs conducted across diverse urban and semi-urban healthcare settings in Ethiopia. Study participants included adults with type 1 and type 2 diabetes, as well as

provided the original author and source are credited.

**Data availability statement:** All relevant data are within the paper and its Supporting Information files.

**Funding:** The author(s) received no specific funding for this work.

healthcare professionals and caregivers. Six major barriers to diabetes self-care were identified: health system limitations, individual-level challenges, socioeconomic constraints, behavioral and lifestyle factors, cultural and social norms, and lack of social support. Key facilitators included strong family and social support, health education and counseling, patient motivation and skills, culturally aligned practices, and signs of health system readiness.

## Conclusion and Recommendation

Diabetes self-care in Ethiopia is shaped by interconnected individual, cultural, and systemic factors. While substantial barriers persist, important enablers, particularly family involvement and supportive healthcare interactions, offer opportunities for improvement. Interventions should be culturally sensitive, community-centered, and supported by strengthened health systems. Policy efforts should prioritize integration of diabetes care into primary healthcare services, capacity building of healthcare providers, and incorporation of patients' lived experiences into care planning. Prospero registration number CRD420251033692

## Introduction

Diabetes mellitus (DM) is a chronic metabolic disorder characterized by elevated blood glucose levels due to insulin deficiency or resistance [1]. Diabetes mellitus is a global health issue with an unprecedented prevalence of 11.1% in adults aged 20–79 years, with over 40% undiagnosed [2]. The number of adults with diabetes is particularly pronounced in low- and middle-income countries (LMICs), where over 80% [3]. Despite this, many individuals in LMICs lack access to adequate diagnosis and treatment, exacerbating the disease's burden [4].

Diabetes mellitus is a growing public health concern in Africa, with 54 million adults aged 20–79 living with the condition in 2022 [5]. Africa has the highest proportion of undiagnosed diabetes cases globally, with over 50% unaware, increasing the risk of complications and mortality [6].

The International Diabetes Federation (IDF) estimated a national adult diabetes prevalence of 3.3% in 2021. Ethiopia's diabetes prevalence accounts for 8% [7], equating to approximately 1.92 million individuals aged 20–79 years living with diabetes in Ethiopia [8]. Inadequate self-care practices and systemic barriers, such as a lack of organized services and limited knowledge, hinder effective diabetes self-care, affecting the healthcare system [9,10].

Effective diabetes management heavily relies on patients' engagement in self-care behaviors, including regular blood glucose monitoring, adherence to medication, dietary modifications, and physical activity [11,12].

In this review, diabetes self-care is understood as the daily, intentional actions undertaken by people living with diabetes to manage their condition, prevent complications, and maintain physical and psychosocial well-being, including medication adherence, dietary regulation, physical activity, blood glucose monitoring, and

appropriate use of healthcare services. Various factors influence individuals' ability to perform these self-care activities, ranging from personal beliefs and knowledge to systemic healthcare challenges [13,14].

In Ethiopia, despite the significance of self-care in diabetes management, many people living with diabetes do not consistently adhere to recommended practices, leading to poor glycemic control, increased complications, and higher healthcare costs, the underlying causes of which are not fully understood [15,16]. Poor diabetes management in Ethiopia imposes a significant burden on individuals and the healthcare system, manifesting in high rates of complications, increased healthcare costs, and diminished quality of life [17–19].

The lack of integrated diabetes care services, including endocrinologists and trained educators, leads to suboptimal diabetes management due to inconsistent availability of tests and medications, patients' lack of understanding, reliance on medications, and lapses in medication adherence [20]. Cultural beliefs, economic constraints, and traditional remedies hinder diabetes self-care practices, leading to patients struggling to afford medications and maintain a diabetes-friendly diet, compromising disease management [21].

Diabetes self-care in Ethiopia can be supported by multidisciplinary care, culturally tailored education, strong social support networks, improved health literacy, and reliable access to essential medications. However, these enabling factors are often undermined by weak implementation, underutilization of social support, low health literacy, and persistent shortages of essential medicines and diagnostic tools, highlighting the need for urgent attention [20–25].

While various studies in Ethiopia have identified barriers and facilitators to diabetes self-care, significant gaps remain. Most focus on individual-level challenges like low awareness and cultural beliefs, with limited attention to broader social determinants, health system factors, and policy-level perspectives. Family support is recognized but not deeply analyzed, and there is a lack of longitudinal and intervention-based research. Therefore, this systematic review aimed to address these gaps by synthesizing diverse evidence, highlighting underexplored areas, comparing contexts, and guiding future interventions and policies with a comprehensive, evidence-based understanding of diabetes self-care practices in Ethiopia.

## Methods

### Study design

This study employed a qualitative systematic review design to synthesize existing research evidence on the barriers and enablers to diabetes self-care among individuals living with diabetes in Ethiopia. The review adhered to the Preferred Reporting Items for Systematic Reviews and Meta-Analyses (PRISMA) 2020 guidelines [26] and the PRISMA Extension for Qualitative Evidence Synthesis (PRISMA-QS) [27], ensuring methodological transparency, rigor, and reproducibility throughout all stages of the review process.

### Aim of the study

This qualitative systematic review aimed to generate a contextualized and comprehensive understanding of the barriers and facilitators influencing diabetes self-care practices in Ethiopia by synthesizing evidence from primary qualitative studies and the qualitative components of mixed-methods studies.

### Review protocol

To ensure accountability and reduce potential bias, a detailed review protocol was developed before commencing the review. The protocol outlined the review objectives, eligibility criteria, search strategy, data extraction procedures, critical appraisal framework, and synthesis method. It was designed by the Joanna Briggs Institute (JBI) Manual for Evidence Synthesis [3], which provides methodological guidance for qualitative reviews. The protocol was registered with PROSPERO under registration number CRD420251033692 on April 01, 2025, serving as a formal commitment to methodological integrity.

## Search strategy

A comprehensive and systematic literature search was conducted across PubMed/MEDLINE, Scopus, CINAHL, Web of Science, and Google Scholar. The search included studies published up to March 15, 2025, and was updated before manuscript submission. Both free-text keywords and Medical Subject Headings (MeSH) terms were used to enhance sensitivity and specificity. Search terms were combined using Boolean operators and included: ("Diabetes" OR "Diabetes Mellitus" OR "Type 2 Diabetes") AND ("Self-care" OR "Self-management") AND ("Barriers" OR "Facilitators" OR "Challenges" OR "Enablers") AND ("Qualitative" OR "Mixed Methods") AND ("Ethiopia"). In addition, backward citation tracking of reference lists was performed to identify any additional eligible studies missed during the initial database searches.

## Inclusion and exclusion criteria

Eligibility criteria were defined using the SPIDER framework, which is well-suited for qualitative evidence synthesis. The Sample (S) included adults diagnosed with diabetes mellitus, as well as healthcare professionals, caregivers, or family members involved in diabetes care within Ethiopia. The Phenomenon of Interest (PI) focused on diabetes self-care practices, specifically barriers and facilitators. The Design (D) included qualitative methodologies such as interviews, focus group discussions, ethnography, and narrative inquiry. Evaluation (E) captured lived experiences, perceptions, beliefs, and contextual influences related to self-care. The Research type (R) included primary qualitative studies and mixed-methods studies with extractable qualitative findings. Studies conducted outside Ethiopia, purely quantitative studies, reviews, editorials, and commentaries were excluded.

## Search and screening process

Search results were imported into Zotero reference management software, and duplicates were removed. Two reviewers (SA and FK) independently screened 100% of all titles and abstracts for eligibility. Full-text articles were retrieved and independently assessed in full by both reviewers. Any discrepancies at either screening stage were resolved through discussion and consensus. The study selection process was documented using the PRISMA 2020 flow diagram, capturing the number of studies identified, screened, excluded, and included, along with reasons for exclusion at each stage (Fig 1). The final database search included studies published up to March 15, 2025. Study selection, data extraction, and synthesis were conducted according to the predefined review protocol and PRISMA 2020 guidelines.

## Data extraction

Data extraction was conducted independently by two reviewers using a standardized, piloted data extraction form. Extracted data included study characteristics, participant details, methodological features, and reported barriers and facilitators to diabetes self-care. Following independent extraction, the reviewers compared extracted data, and any discrepancies or inconsistencies were resolved through discussion and consensus to ensure accuracy and completeness. When study information was missing or unclear, interpretation was restricted to the data explicitly reported in the published articles, and no assumptions were made beyond the available information. Authors were not contacted for additional clarification, as all included studies provided sufficient qualitative data to support thematic synthesis.

## Researcher reflexivity

To enhance the credibility and trustworthiness of this qualitative synthesis, reflexivity was actively maintained throughout the review process. The reviewers brought diverse professional backgrounds in nursing, public health, and qualitative research, which offered valuable perspectives but also necessitated careful attention to potential biases. Reflexive practices, including journaling, memo writing, and regular team discussions, were employed to critically examine how personal assumptions, prior experiences with diabetes care, and contextual knowledge of the Ethiopian healthcare system

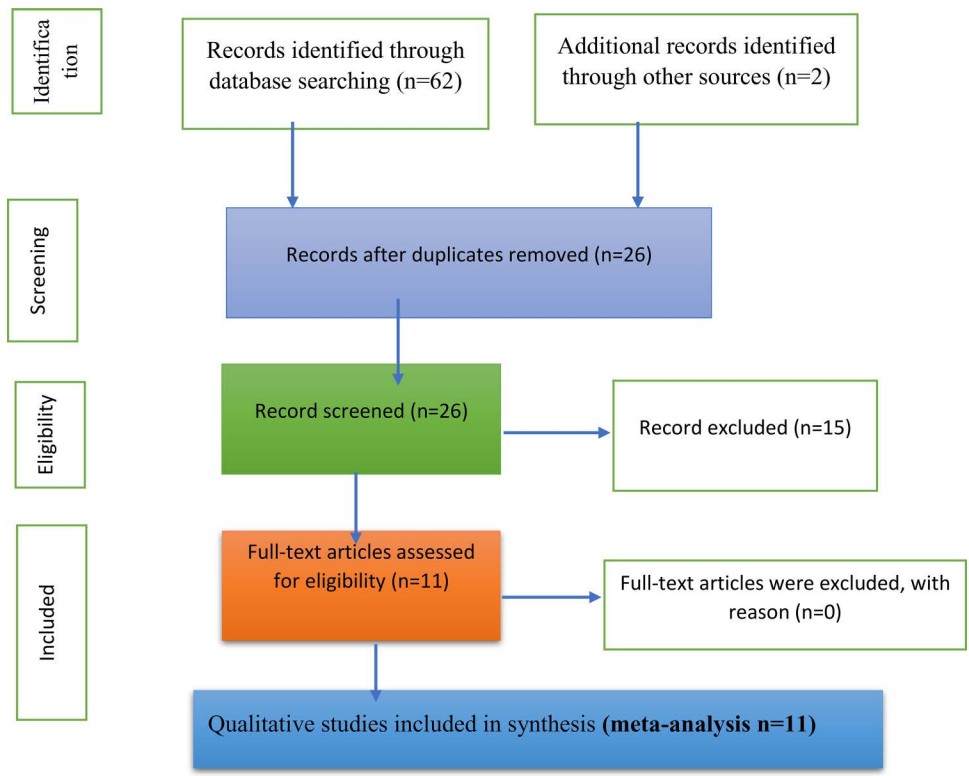

**Fig 1. PRISMA 2020 flow diagram describing the selection of studies for the systematic review and meta-analysis on barriers and enablers on diabetic self-care practice in Ethiopia, 2025.**

might shape data interpretation. By consciously acknowledging and interrogating our positionalities, we sought to remain transparent and minimize subjective influence during data extraction, coding, and thematic synthesis. This reflexive stance strengthened the integrity and rigor of the review process.

## Data synthesis

Thematic synthesis, as described by Thomas and Harden [28], was employed to integrate and interpret findings across studies. Initial line-by-line coding was performed independently by both reviewers. Codes were then compared and collaboratively refined into descriptive and analytical themes. This method involved: (1) line-by-line coding of extracted findings, including both participant quotes and author interpretations; (2) organizing codes into descriptive themes; and (3) developing analytical themes that extended beyond individual study findings to offer new insights into diabetes self-care in Ethiopia. The synthesis process was iterative and collaborative, with continuous engagement among reviewers to refine themes and ensure alignment with the review question and Ethiopian context.

## Quality appraisal

To assess the methodological quality of the included studies, the Critical Appraisal Skills Programme (CASP) Qualitative Checklist was applied [29]. This tool examines aspects such as clarity of the research aim, appropriateness of design, sampling strategies, data collection, ethical considerations, reflexivity, and analytical rigor. Two reviewers independently appraised all included studies using the CASP Qualitative Checklist. Disagreements were resolved through discussion,

with consensus reached in all cases. Although no study was excluded based on quality, the appraisal helped inform the interpretation and trustworthiness of the review findings.

## Assessment of confidence in the evidence

The GRADE-CERQual (Confidence in the Evidence from Reviews of Qualitative Research) [30] approach was used to assess the confidence in each synthesized theme. This framework evaluates the methodological limitations of the contributing studies, coherence of the data, adequacy (richness and quantity), and relevance to the review question. Each key finding was assigned a confidence level (high, moderate, low, or very low), and justifications were documented to ensure transparency.

## Ethical considerations

This review did not involve the collection of primary data and was based exclusively on previously published studies. Therefore, formal ethical approval was not required. Nonetheless, ethical rigor was ensured by including only studies that had received ethical clearance from their respective institutional review boards and had obtained informed consent from all participants. By synthesizing ethically conducted research, this review upholds the principles of research integrity and respect for participants.

# Result

## Study selection

A total of 62 records were retrieved from the database search and 02 additional results through other sources, of which 64 unique records were obtained. After screening titles and abstracts, 26 articles remained for full-text screening. Eleven of these studies met the eligibility criteria (Fig 1). Studies excluded after full-text review (n = 15) and their reasons for exclusion are provided in Supplementary file (S1 File). The included studies, published between 2013 and 2023, comprised a diverse range of qualitative and mixed-methods research conducted across Ethiopia to explore the multifaceted barriers and facilitators influencing diabetes self-care [21–23,25,31–37]. These studies were situated in various geographic and institutional contexts, encompassing both urban and semi-urban health facilities, such as Tikur Anbessa Hospital in Addis Ababa [21,33], Dessie Referral Hospital in Northern Ethiopia [36,37], and Kemisse General Hospital in the northeast [22].

Methodologically, most studies adopted robust qualitative approaches, including descriptive [25], phenomenological [22,31], and exploratory designs [21], while others integrated mixed methods to capture both quantitative and qualitative dimensions [34,36]. This methodological pluralism allowed researchers to explore both lived experiences and broader patterns of self-care adherence, guided by theoretical frameworks such as the Health Belief Model and Theory of Planned Behavior [34].

The sample populations were also diverse, ranging from people living with type 1 diabetes as young as 13 years old [22] to older adults living with type 2 diabetes with long disease durations [31]. Several studies also incorporated key informants, including healthcare professionals and caregivers, reflecting a multidimensional understanding of diabetes care ecosystems [23,25]. Data collection techniques primarily relied on in-depth and semi-structured interviews, often supported by thematic analysis using software such as Atlas. Ti or NVivo [25,33], while some studies also employed structured questionnaires and direct observation to enrich data validity [22,36] (Table 1).

## Methodological quality of included studies

The overall methodological quality of the included studies was robust, demonstrating rigorous design and thoughtful execution across diverse qualitative paradigms. All eleven studies clearly articulated their research aims and employed appropriate qualitative methodologies aligned with their objectives, including phenomenological, descriptive, exploratory, and mixed-methods approaches [21–23,25,31–37]. Most studies adopted sound recruitment strategies and collected data

**Table 1. Charectrstic of included study on exploring barriers and enablers to diabetes self-care in Ethiopia, 2025.**

| Author(s) & Year | Study Setting | Research Objectives | Study Design | Participant Characteristics | Data Collection & Analysis Methods | Key Findings (Barriers) & Illustrative Participant Quotes | Key Findings (Facilitators) & Illustrative Participant Quotes |
|---|---|---|---|---|---|---|---|
| Berhe et al., 2022 | Tigray Region (Ayder, Mekelle, Quiha Hospitals) | To explore barriers to self-care management among type 2 diabetes patients | Descriptive qualitative study | 22 T2DM patients, 10 caregivers, 10 HCPs | In-depth interviews; Thematic analysis using Atlas.ti | - Lack of education: "Not well informed about diabetes."- Psychological distress: "I feel hopeless and tired… it's too much."- Poor lifestyle adherence: "It's hard to change what I've done for years."- No social support: "I live alone… there's no one to remind me."- Health system gaps: "We wait long and can't get medication." | - Family support: "My daughter helps me remember to take meds."- Tailored advice from HCPs: "They show me how to eat and walk." |
| Bogale et al., 2022 | Kemisse General Hospital (NE Ethiopia) | To explore lived experiences and self-care practice of Type 1 DM | Phenomenological qualitative study | 13 T1DM patients (ages 13–70, mix of sex, education) | In-depth interviews + observation; Thematic analysis with Atlas.ti | - Medication barriers: "I skipped insulin because I was traveling."- Lack of foot care & diet adherence: "I didn't know I had to check my feet."- No glucometer: "I can't afford strips. I guess based on how I feel." | - Family role: "My mom gives me injections when I feel scared."- Practical skills: "I learned how to rotate injection sites." |
| Endalew Hailu, 2014 | Dilla University Referral Hospital | To assess perception on diabetes and self-care practices | Cross-sectional with qualitative supplement | 310 diabetic patients (quantitative); 10 for qualitative (mixed type 1 & 2) | Questionnaire + in-depth interviews; descriptive stats + thematic analysis | - Inconsistent self-monitoring: "I don't check my blood sugar, no machine."- Poor physical activity: "I just don't have time to walk."- Misunderstanding about disease: "Some say it goes away with prayer." | - Education level: "Because I went to school, I know what not to eat."- Positive attitude: "I take my meds because I believe it helps." |
| Habte et al., 2017 | Tikur Anbessa, Yekatit 12 (Addis Ababa), Butajira Hospital | To identify barriers and facilitators to adherence to anti-diabetic medications | Qualitative: In-depth interviews | 39 adults with T2DM (≥1 year on treatment) from diverse socio-demographics | Semi-structured interviews, open coding using NVivo | Barriers: Illness perceptions (e.g., "Diabetes leads to many different complications…"), side effects, religious beliefs, poor provider interaction, medication unavailability- Quote: "I abandoned it for 8 months [in favor of holy water]... I became so ugly looking." | Facilitators: Perception of severity, social support, seeing results of treatmentQuote: "The medicine would never be discontinued… even if there is fasting." |
| Letta et al., 2022 | Harar, Eastern Ethiopia (2 public hospitals) | To explore barriers to diabetes self-care practices | Mixed-ethods (Qualitative: HCP interviews) | 26 healthcare providers across disciplines | In-depth interviews; thematic analysis with ATLAS.ti | Barriers: Poor diabetes knowledge, low literacy, cultural beliefs, food insecurity, inadequate staff and training, irregular medication suppliesQuote: "We often don't have glucometers or strips to give them... they can't monitor at home." | Facilitators: Family support, consistent health education, improved access to tools and trained staffQuote: "When we educate both the patient and their caregiver, we see better outcomes." |

*(Continued)*

| Author(s) & Year | Study Setting | Research Objectives | Study Design | Participant Characteristics | Data Collection & Analysis Methods | Key Findings (Barriers) & Illustrative Participant Quotes | Key Findings (Facilitators) & Illustrative Participant Quotes |
|---|---|---|---|---|---|---|---|
| Simegn et al., 2023 | Comprehensive Specialized Hospitals (Tibebe Gion, Debre Tabor, Dessie) | To assess adherence to self-care practices using TPB & HBM | Mixed survey + phenomenological qualitative | 846 T2DM patients (quant), 16 key informants (qual) | SDSCA survey; qualitative interviews with thematic analysis | Barriers: Poor behavioral control, low perceived benefit, financial constraints, limited self-efficacyQuote: "I know what to do, but when I'm tired or broke, I just skip the routine." | Facilitators: Diabetes association membership, glucometer ownership, good perceived self-efficacyQuote: "Having my own machine helps me stay on track with checking sugar." |
| Tewahido & Berhane (2017) | Addis Ababa, Ethiopia (Menelik II and Zewditu Memorial Hospitals) | To describe diabetes self-care practices and identify facilitators and barriers among Type 2 diabetes patients. | Qualitative study using in-depth interviews | 13 participants, aged 35–65, Type 2 DM for ≥5 years | Semi-structured interviews; thematic analysis using OpenCode | Barriers: Irregular blood sugar monitoring, poor dietary adherence, limited physical activity, cultural/social pressures, financial constraints. "I eat what I like and don't want to be picky…" "It's not comfortable to go around with injection equipment." | Facilitators: Better medication adherence. "I never omit my medication on purpose." Participants categorized as "compliant", "confused", or "negligent." |
| Tuha et al. (2021) | Dessie Referral Hospital, NE Ethiopia | To assess knowledge and practice of diabetic foot self-care and associated factors. | Mixed (cross-sectional survey + phenomenological interviews) | 344 participants (quantitative), 12 key informants (qualitative) | Structured questionnaire and in-depth interviews; content analysis | Barriers: Low knowledge, infrequent foot inspection, lack of equipment (e.g., glucometer), social/cultural norms. "I don't have a home sugar meter… I go to the clinic when I feel ill." "Saying that my feet are fine; I did not always see my feet." | Facilitators: Religious practices, good awareness among some patients, support from health professionals. "I wash my feet morning and night... up to five times a day." |
| Zewdie et al. (2022) | Dessie Referral Hospital, Northern Ethiopia | To assess self-care practice and associated factors among Type 2 DM patients. | Mixed-method (cross-sectional + in-depth interviews) | 328 participants; 13 for qualitative interviews | Exit interviews; SDSCA questionnaire; thematic analysis | Barriers: Age, co-morbidities, lack of education, poor access, lack of social support. "There are no fruits and vegetables in our village." "No one reminds me to take my medication." | Facilitators: Diabetes education, counseling, social support. "I got advice from my nurse on how to take care of myself." |

*(Continued)*

**Table 1.** (Continued)

| Author(s) & Year | Study Setting | Research Objectives | Study Design | Participant Characteristics | Data Collection & Analysis Methods | Key Findings (Barriers) & Illustrative Participant Quotes | Key Findings (Facilitators) & Illustrative Participant Quotes |
|---|---|---|---|---|---|---|---|
| Tigestu Alemu Desse et al., 2022 | Diabetes Centre, Tikur Anbessa Specialised Hospital (TASH), Ethiopia | To examine perspectives of patients, health professionals, and policymakers on current practices and future preferences for type 2 diabetes care in a tertiary hospital in Ethiopia | Exploratory qualitative study | 59 participants: patients (n = 30), health professionals and policymakers (n = 29); 48 interviews and 11 focus group participants | Semi-structured interviews and focus group discussions; thematic analysis | Barriers:• Lack of essential resources: medications, lab and diagnostics, and diabetes educators• Infrequent monitoring and follow-up• Absence of structured diabetes education• Poor infrastructure and physical setup compromising privacy- Quote: "We don't get all medicines we need from the hospital… even the test results take long… it is discouraging." | Facilitators:• Availability of endocrinologists and trained nurses for foot and eye care• Stakeholders showed interest in developing tailored, patient-centred care models• Desire for structured education and reliable follow-upQuote: "Having a proper system that gives us education and regular follow-ups would help manage our diabetes better." |
| Bayked et al. - 2022 | North-East Ethiopia (likely Dessie) | To examine type 2 diabetes patients' perceptions of the causes of their illness | Qualitative phenomenological study with modified grounded theory, based on the causal dimension of the Common-Sense Model (CSM) | 24 participants (11 males, 13 females); aged 35–75; median age 57; average diabetes duration 12 years | Semi-structured face-to-face interviews; purposive sampling; analysis with QDA Miner Lite v2.0.8; thematic analysis | Barriers:• Misattribution of diabetes to emotional distress, supernatural causes, and misfortune• Beliefs in psycho-economic or spiritual causes may delay biomedical treatment-seekingQuote: "It came after I lost my job… I believe being cursed or being a victim of evil eyes led to this illness." | Facilitators:• Recognition of stress and lifestyle as possible contributing factors• Cultural narratives that can be engaged for education and tailored interventionsQuote: "I think too much thinking and worry caused this, maybe if I manage stress, it can help." |

through in-depth or semi-structured interviews, often complemented by observational methods or supplementary surveys [22,25,34].

Data analysis procedures were consistently rigorous, utilizing thematic analysis supported by qualitative software such as Atlas. Ti, NVivo, and Open Code [21–23,25,31–37]. Findings were presented and well-supported with illustrative quotes, enhancing transparency and trustworthiness. However, a common methodological limitation was the limited reporting on the relationship between researchers and participants. While some studies partially acknowledged reflexivity, most did not fully address how researcher positionality might have influenced data collection or interpretation. Furthermore, ethical considerations were not consistently detailed, with many studies omitting information about consent procedures or ethics approvals. Despite these gaps, the studies were judged to have high to moderate confidence in their contributions, with nearly all rated as "highly valuable" due to their contextual relevance, analytic depth, and practical implications for diabetes self-care in Ethiopia [21,23,33]. These findings affirm the methodological soundness of the current evidence base, while also highlighting areas for improvement in future qualitative health research within similar contexts (Table 2).

## Synthesis of barriers to diabetes self-care practice in Ethiopia

A synthesis of qualitative findings revealed six interrelated themes that constitute significant barriers to effective diabetes self-care in Ethiopia: health system limitations, individual-level challenges, socioeconomic constraints, behavioral and lifestyle factors, cultural and social norms, and lack of social support (Table 3).

Table 2. Critical appraisal skills programme checklist for qualitative research quality appraisal on exploring barriers and enablers to diabetes self-care in Ethiopia, 2025.

| Author and year | 1. Was there a clear statement of the aims of the research? | 2. Is a qualitative methodology appropriate? | 3. Was the research design appropriate to address the aims of the research? | 4. Was the recruitment strategy appropriate to the aims of the research? | 5. Was the data collected in a way that addressed the research issue? | 6. Has the relationship between the researcher and participants been adequately considered? | 7. Have ethical issues been taken into consideration? | 8. Was the data analysis sufficiently rigorous? | 9. Is there a clear statement of findings? | 10. How valuable is the research? |
|---|---|---|---|---|---|---|---|---|---|---|
| Berhe et al., 2022 | Yes | Yes | Yes | Yes | Yes | Partially considered | Not stated | Yes | Yes | High |
| Bogale et al., 2022 | Yes | Yes | Yes | Yes | Yes | Partially considered | Not stated | Yes | Yes | High |
| Endalew Hailu, 2014 | Yes | Yes | Yes | Yes | Yes | Not stated | Partially stated | Yes | Yes | Moderate–High |
| Habte et al., 2017 | Yes | Yes | Yes | Yes | Yes | Partially considered | Not stated | Yes | Yes | High |
| Letta et al., 2022 | Yes | Yes | Yes | Yes | Yes | Partially considered | Not stated | Yes | Yes | High |
| Simegn et al., 2023 | Yes | Yes | Yes | Yes | Yes | Partially considered | Not stated | Yes | Yes | High |
| Tewahido & Berhane, 2017 | Yes | Yes | Yes | Yes | Yes | Partially considered | Not stated | Yes | Yes | High |
| Tuha et al., 2021 | Yes | Yes | Yes | Yes | Yes | Partially considered | Not stated | Yes | Yes | High |
| Zewdie et al., 2022 | Yes | Yes | Yes | Yes | Yes | Partially considered | Not stated | Yes | Yes | High |
| Tigestu Alemu Desse et al., 2022 | Yes | Yes | Yes | Yes | Yes | Partially considered | Not stated | Yes | Yes | High |
| Bayked et al., 2022 | Yes | Yes | Yes | Yes | Yes | Partially considered | Not stated | Yes | Yes | High |

## Health system factors

Health system factors were among the most frequently cited barriers across the studies [21,23,25,33]. Patients reported challenges such as medication shortages, lack of glucometers and test strips, and insufficient staffing, which led to delays in diagnosis, poor follow-up, and missed appointments, "We wait long and can't get medication" [23,25], while another emphasized the frustration of inconsistent care, stating, "We don't get all medicines we need from the hospital… even the test results take long… it is discouraging" [21]. In some instances, poor provider interaction caused patients to abandon conventional treatment; as the participant explained, "I abandoned it for 8 months [in favor of holy water]... I became so ugly looking" [33].

Table.3. Selected quotes from studies on barriers to diabetes self-care in Ethiopia: Themes, subthemes, and illustrative quotes.

| Theme | Subtheme | Illustrative Quote | Contributing Studies |
|---|---|---|---|
| Health System Factors | Medication/ Resource Shortages | "We wait long and can't get medication." | Berhe et al., 2022 |
| | Poor infrastructure and follow-up | "We don't get all medicines we need from the hospital… even the test results take long… it is discouraging." | Tigestu Alemu Desse et al., 2022 |
| | Inadequate staffing/ training | "We often don't have glucometers or strips to give them... they can't monitor at home." | Letta et al., 2022 |
| | Poor provider interaction | "I abandoned it for 8 months [in favor of holy water]... I became so ugly looking." | Habte et al., 2017 |
| Individual level Factors | Psychological distress | "I feel hopeless and tired… it's too much." | Berhe et al., 2022 |
| | Limited self-efficacy | "I know what to do, but when I'm tired or broke, I just skip the routine." | Simegn et al., 2023 |
| | Lack of awareness and skills | "I didn't know I had to check my feet." | Bogale et al., 2022 |
| | Misconceptions and beliefs | "Some say it goes away with prayer." | Endalew Hailu, 2014 |
| | Spiritual/attribution beliefs | "I believe being cursed or being a victim of evil eyes led to this illness." | Bayked et al., 2022 |
| Socioeconomic Barriers | Financial constraints | "I can't afford strips. I guess based on how I feel." | Bogale et al., 2022 |
| | Food insecurity | "There are no fruits and vegetables in our village." | Zewdie et al., 2022 |
| Lifestyle & Behavior | Poor lifestyle adherence | "It's hard to change what I've done for years." | Berhe et al., 2022 |
| | Poor physical activity | "I just don't have time to walk." | Endalew Hailu, 2014 |
| | Irregular monitoring | "I don't check my blood sugar, no machine." | Endalew Hailu, 2014; Tewahido & Berhane, 2017 |
| Cultural & Social Norms | Religious practices interfering with care | "The medicine would never be discontinued… even if there is fasting." | Habte et al., 2017 |
| | Cultural/social stigma or pressure | "It's not comfortable to go around with injection equipment." | Tewahido & Berhane, 2017 |
| Lack of Support | Social isolation | "I live alone… there's no one to remind me." | Berhe et al., 2022 |
| | Absence of reminders | "No one reminds me to take my medication." | Zewdie et al., 2022 |

## Individual-level factors

At the individual level, psychological distress, limited self-efficacy, and poor disease knowledge significantly impede self-care. Patients reported feeling overwhelmed, confused, or incapable of sustaining the required lifestyle changes. Misconceptions such as viewing diabetes as curable through prayer or attributing the condition to supernatural causes further delayed timely medical care [31,32]. At the individual level, several studies highlighted the burden of psychological distress, low confidence in self-management, and limited understanding of the disease [22,25,31,32,34]. Patients frequently described emotional fatigue and demotivation, with one stating, "I feel hopeless and tired… it's too much" [22,25]. Limited self-efficacy was also a concern, as expressed by a participant who said, "I know what to do, but when I'm tired or broke, I just skip the routine" [34]. Misconceptions and spiritual beliefs about the disease further undermined effective care-seeking. One participant noted, "Some say it goes away with prayer" [32], while another remarked, "I believe being cursed or being a victim of evil eyes led to this illness" [31].

## Socioeconomic barriers

Socioeconomic constraints were another common theme, particularly in rural or low-income areas [22,37]. Patients frequently cited financial hardship as a reason for irregular monitoring and dietary non-adherence. For example, one participant explained, "I can't afford strips. I guess based on how I feel" [22], while another shared, "There are no fruits and vegetables in our village" [37].

## Behavioral and lifestyle factors

The theme of behavioral and lifestyle factors emerged through participants' struggles with adapting to new habits [25,32,35]. Long-standing behaviors and time constraints were major challenges. One patient reflected, "It's hard to change what I've done for years" [25], and another explained, "I just don't have time to walk" [32]. The inability to monitor blood sugar regularly was also cited: "I don't check my blood sugar, no machine" [32,35].

## Cultural and social norms

Cultural and social norms also shape adherence behaviors. For instance, fasting practices and social stigma around insulin injections posed unique challenges [33,35]. One participant described how faith practices influenced adherence: "The medicine would never be discontinued… even if there is fasting" [33]. Another patient revealed, "It's not comfortable to go around with injection equipment" [35], pointing to stigma and embarrassment in public settings.

## Lack of social support

Finally, a lack of social support was closely linked to poor adherence and emotional fatigue [25,37]. Patients living alone or without caregiver assistance often forgot medications or felt unmotivated. "I live alone… there's no one to remind me," said one participant [25], while another noted, "No one reminds me to take my medication" [37], underscoring the importance of interpersonal support in chronic disease management.

## Synthesis of facilitators to diabetes self-care practice in Ethiopia

The synthesis of qualitative evidence also identified five overarching themes that facilitate effective diabetes self-care among patients in Ethiopia: social and family support, health system enablers, individual capacities, religious and cultural strengths, and systemic readiness (Table 4).

## Social and family support

One of the most consistently reported facilitators was social and family support [22,23,25,34]." Patients emphasized the vital role played by close family members and caregivers in helping them manage medication schedules, diet, and emotional well-being. For example, one patient noted, "My daughter helps me remember to take meds" [25], while another said, "My mom gives me injections when I feel scared" [22]. Healthcare professionals also highlighted that When we educate both the patient and their caregiver, we see better outcomes" [23]. Moreover, community support structures such as diabetes associations offered encouragement and shared learning opportunities. As one participant shared, "Diabetes association membership…" helped them stay motivated and informed [34].

## Health system enablers

Several studies pointed to health system enablers, particularly in the form of targeted education and counseling. Participants who received individualized instruction and consistent follow-up were more likely to adhere to self-care routines [21,25,37]. One remarked, "They show me how to eat and walk" [25], while another reported, "I got advice from my nurse on how to take care of myself" [37]. The availability of trained personnel, including diabetes-focused nurses and educators, also proved valuable. "Having a proper system that gives us education and regular follow-ups would help…" [21].

**Table 4. Selected quotes from studies on facilitators to diabetes self-care in Ethiopia: Themes, subthemes, and illustrative quotes.**

| Theme | Subtheme | Illustrative Quote | Contributing Studies |
|---|---|---|---|
| **Social Support** | Family involvement | "My daughter helps me remember to take meds." | Berhe et al., 2022 |
| | Caregiver participation | "My mom gives me injections when I feel scared." | Bogale et al., 2022 |
| | Social encouragement | "When we educate both the patient and their caregiver, we see better outcomes." | Letta et al., 2022 |
| | Peer and community support | "Diabetes association membership…" | Simegn et al., 2023 |
| **Health System Enablers** | Health education | "They show me how to eat and walk." | Berhe et al., 2022 |
| | Consistent counseling | "I got advice from my nurse on how to take care of myself." | Zewdie et al., 2022 |
| | Availability of trained personnel | "Having a proper system that gives us education and regular follow-ups would help…" | Tigestu Alemu Desse et al., 2022 |
| **Individual Capacities** | Self-efficacy | "Having my own machine helps me stay on track with checking sugar." | Simegn et al., 2023 |
| | Health knowledge | "Because I went to school, I know what not to eat." | Endalew Hailu, 2014 |
| | Practical diabetes management skills | "I learned how to rotate injection sites." | Bogale et al., 2022 |
| | Positive attitude | "I take my meds because I believe it helps." | Endalew Hailu, 2014 |
| **Religious & Cultural** | Supportive religious practices | "I wash my feet morning and night... up to five times a day." | Tuha et al., 2021 |
| | Cultural understanding of stress | "Maybe if I manage stress, it can help." | Bayked et al., 2022 |
| **Systemic Readiness** | Policy and professional interest | "Stakeholders showed interest in developing tailored, patient-centred care models." | Tigestu Alemu Desse et al., 2022 |
| | Access to basic tools and medications | "Improved access to tools and trained staff." | Letta et al., 2022 |
| | Regular follow-up | "Having a proper system that gives us education and regular follow-ups would help…" | Tigestu Alemu Desse et al., 2022 |

### Individual capacities

Individual capacities such as health knowledge, confidence, and practical skills were crucial for enabling self-care [22,32,34]. Patients who had a strong educational background or had received adequate training were more likely to understand and implement recommended practices. "Because I went to school, I know what not to eat," said one participant [32]. Others emphasized their confidence and capability in daily management tasks: "Having my machine helps me stay on track with checking sugar" [34], and "I learned how to rotate injection sites" [22]. Positive beliefs and motivation were also pivotal, as expressed by a participant who stated, "I take my meds because I believe it helps" [32].

### Religious and cultural practices

Interestingly, some studies found that religious and cultural practices, often perceived as barriers, could also serve as facilitators when aligned with self-care behaviors [31,36]. For instance, a patient shared, "I wash my feet morning and night... up to five times a day" [36], reflecting how religious hygiene routines complemented foot care recommendations. Another participant reflected on the psychological value of managing stress, stating, "Maybe if I manage stress, it can help" [31], linking traditional narratives with modern coping strategies.

### Signs of systemic readiness

Signs of systemic readiness also emerged, especially in tertiary settings where specialized care and policy attention were evident [21,23]. One participant shared hope for institutional improvements: "Stakeholders showed interest in developing tailored, patient-centered care models" [21]. Enhanced access to medical tools and better coordination of services were also mentioned as critical supports: "Improved access to tools and trained staff" made a notable difference [23].

## Confidence in review findings (GRADE-CERQual assessment) of themes

The study explored barriers and enablers to diabetes self-care in Ethiopia using the GRADE-CERQual approach to assess confidence in synthesized themes. High-confidence evidence highlighted systemic challenges such as medication shortages, poor infrastructure, and workforce gaps as key barriers. Psychosocial distress, limited disease knowledge, and low self-efficacy also hindered self-care, supported by moderate to high confidence. Socioeconomic constraints and cultural beliefs presented additional challenges, while family and social support emerged as strong enablers. Education, motivation, and practical skills significantly promoted self-care, with high confidence. Lastly, the need for integrated, patient-centered care models was strongly endorsed, marking a vital step for future diabetes care reform in Ethiopia (Table 5).

## Discussion

This qualitative systematic review synthesized evidence from 11 studies to explore barriers and enablers of diabetes self-care practice in Ethiopia. The findings demonstrate that self-care behaviors are shaped by a complex interaction of individual, social, cultural, and health system factors. While previous studies have documented isolated challenges, this synthesis provides an integrated understanding of how these factors intersect to influence diabetes self-management in resource-limited settings. This review examines barriers and enablers to diabetes self-care in Ethiopia, across various healthcare settings [21–23,25,31–37]. The studies used qualitative and mixed-methods designs, including descriptive, phenomenological, and exploratory approaches. The involvement of patients along with healthcare professionals and caregivers in the studies underscores the importance of social support in facilitating self-care, providing a comprehensive understanding of patients' experiences. The synthesis of qualitative findings from this review revealed six major barriers to effective diabetes self-care in Ethiopia [21–23,25,31–37]: health system limitations, individual-level challenges, socio-economic constraints, behavioral and lifestyle factors, cultural and social norms, and lack of social support. These barriers highlight the complex and multifaceted challenges that individuals with diabetes face in Ethiopia, and they provide valuable insights for understanding the broader context of diabetes self-management in low-resource settings.

Health system limitations, including inconsistent availability of medications, diagnostic tools, and trained personnel, emerged as dominant barriers and continue to undermine continuity of care. [21,23,25,33]. Patients reported challenges such as including inconsistent access to medications, lack of glucometers and test strips, insufficient staffing, and limited diabetes education programs, which led to delays in diagnosis, poor follow-up, and missed appointments [23,25], while another emphasized the frustration of inconsistent care [21] and poor provider interaction [33]. This finding is consistent with evidence from studies conducted in Ethiopia [38,39], Kenya [40], India [41], Nigeria [42,43], Bangladesh [44,45], Pakistan [46,47]. This is due to Underfunded healthcare systems, inadequate infrastructure, a shortage of trained professionals, and inadequate supplies contributing to health system limitations. The governments should increase funding for non-communicable disease programs, ensure consistent medication supply, train healthcare professionals, integrate diabetes management, and enhance patient-provider communication.

Health system barriers in Ethiopia hinder diabetes self-care, while high-income countries such as the UK and Australia [48,49] facilitate better management through consistent medication access, structured education, and integrated healthcare. These advantages stem from superior health financing, universal coverage, and continuous workforce education, highlighting disparities in diabetes management effectiveness.

Individual-level challenges are another theme of barriers, as Patients often struggle with limited knowledge about their condition, misconceptions about diabetes, psychological distress, limited self-efficacy, and feelings of hopelessness or denial [22,25,31,32,34]. Similar findings were reported in Ethiopia [50,51], Nigeria [42], Bangladesh [45], India [41], and Pakistan [47], where low health literacy and poor understanding and misconceptions about diabetes management were barriers to self-care. This is due to individual diabetes challenges stemming from low health literacy, cultural beliefs, stigma, and insufficient psychosocial support. Targeted interventions like community awareness programs can improve

**Table 5. GRADE-CERQual summary of qualitative findings exploring barriers and enablers to diabetes self-care in Ethiopia, 2025.**

| Theme | Methodological Limitations | Coherence | Adequacy | Relevance | Confidence Level | Rationale |
|---|---|---|---|---|---|---|
| Health System Challenges | Low: Studies used strong designs with triangulation. | High: Clear convergence on medication shortages, staffing gaps, and weak infrastructure. | High: Multiple quotes from patients, caregivers, and HCPs across regions. | High: Strong alignment with public healthcare conditions in Ethiopia. | High | Findings were consistently reported across varied regions and participants; clearly reflect real-world constraints in service delivery and continuity of care. |
| Psychosocial & Emotional Barriers | Low to Moderate: Some variation in depth across studies. | High: Strong consistency in reporting distress and hopelessness. | Moderate: Rich quotes but from a narrower demographic. | High: Mental state directly impacts self-care behavior. | Moderate–High | Emotional distress was a persistent theme, though some studies lacked deeper psychological probing or formal mental health assessment. |
| Knowledge and Self-Efficacy Gaps | Low: Strong thematic rigor in most studies. | High: Consistent reports of limited disease understanding and self-efficacy. | High: Clear narrative presence and patient voice. | High: Directly relevant to managing chronic conditions. | High | Participants repeatedly linked poor outcomes to knowledge gaps and low confidence in disease management—well captured in diverse settings. |
| Socioeconomic and Material Constraints | Low: Robust qualitative methods employed across contexts. | High: Uniform findings on poverty, diet limitations, and supply inaccessibility. | Moderate: Strong quotes, though fewer focused studies. | High: Highly relevant in resource-constrained environments. | Moderate–High | Widespread but not always the central focus; still an essential and well-articulated barrier to regular self-care. |
| Cultural and Spiritual Beliefs | Low to Moderate: Well-conducted, though depth varies. | Moderate: Themes are recurrent but interpretation differs. | Moderate: Select quotes show strong influence on care-seeking. | High: Culturally grounded beliefs shape health behavior. | Moderate | Important insights, especially on religious substitutes for biomedical care, but representation from some regions or groups remains limited. |
| Family and Social Support | Low: Clear and methodologically sound across studies. | High: Recurrent emphasis on caregiver involvement and reminders. | High: Rich quotes from patients, families, and providers. | High: Key enabler of adherence and emotional resilience. | High | Strong evidence showing consistent role of social support as a facilitator across age, sex, and setting. |
| Healthcare Education & Guidance | Low: Most studies had structured interviews and coding. | High: Strong alignment on the importance of tailored counseling. | High: Saturated with quotes on effective provider communication. | High: Core component of diabetes self-management. | High | Robust qualitative data confirms that education empowers both patients and caregivers, directly influencing behavior change. |
| Patient-Level Motivation and Skills | Low: Strong descriptive findings and participant diversity. | High: Clear trends in positive attitude and practical skill uptake. | High: Representative of diverse lived experiences. | High: Crucial to sustainable chronic care. | High | Repeatedly emphasized in multiple studies with illustrative examples—practical, emotional, and motivational facilitators clearly described. |
| Religious and Cultural Enablers | Moderate: Less in-depth probing in some accounts. | Moderate: Spiritual framing aids adherence for some, not all. | Moderate: Helpful but not as richly detailed as other themes. | High: Religion often intersects with health choices. | Moderate | Not universally reported, but when present, cultural/religious frameworks were useful in framing positive behavior and could be leveraged for interventions. |
| System Readiness and Innovation | Low to Moderate: In-depth input from HCPs and stakeholders. | High: Strong convergence on need for structured, patient-centered models. | Moderate–High: Less patient voice, but strong institutional data. | High: Tied to future planning and service quality. | High | Strong professional consensus on gaps and solutions; reflects real health system needs and aspirations—critical for strategic improvements. |

patient education and treatment adherence. In contrast, structured diabetes education effectively addresses barriers in high-income areas, with systematic reviews showing that low health literacy inhibits self-care [52]. Culturally adapted services enhance self-efficacy. Conversely, Ethiopia highlights the urgent need for improved health communication and education programs in developed countries to better manage diabetes.

Socioeconomic constraints were another common theme, particularly in rural or low-income areas. Patients frequently cited financial hardship as a reason for irregular monitoring and dietary non-adherence [22,37]. These challenges mirror barriers documented in Ethiopia [53], South Africa [54], India [41], and Nepal [55], where economic constraints were found

to directly influence patients' ability to purchase medications or follow dietary advice. Governments should implement policies like subsidizing medications, expanding insurance, and integrating diabetes care.

The theme of behavioral and lifestyle factors emerged through participants' struggles with adapting to new habits, including long-standing behaviors, time constraints, and inability to regularly monitor blood sugar due to lack of a machine [25,32,35]. This is consistent with studies done in Ethiopia [53,56], Egypt [57], and the USA [58]. Behavioral and lifestyle challenges in people living with diabetes often stem from rooted habits, limited motivation, lack of support, and competing life demands. Healthcare systems should prioritize patient-centered interventions, including counseling, regular follow-ups, and behavior change support, while community health workers and peer educators can reinforce healthy habits.

Cultural and social norms also shaped adherence behaviors as Stigma, reliance on traditional medicine, fasting practices, and cultural misconceptions around ant diabetic medication were commonly cited by Ethiopian patients [33,35]. The finding similarly reported in study done in Ethiopia [59], Nigeria [42],Bangladesh [45], India [41],and Pakistan [47]. Addressing these challenges requires culturally sensitive health education, community leaders, religious figures, traditional healers, public awareness campaigns, and peer-led support groups. In high-income countries [60], diabetes self-care remains largely unaffected by cultural norms due to supportive programs. Conversely, Ethiopia may benefit from effective health communication to address cultural barriers. The absence of tailored programs in low-resource settings amplifies the influence of cultural norms on diabetes management.

A recurring theme in the Ethiopian context was the absence of supportive social networks, including family and peer support, which adversely affected patients' motivation and capacity for self-care [25,37]. This is in line with evidence from South Africa [61], Saudi Arabia [58], Egypt [57], Mexico [62], and the USA [63], where family involvement and support have been shown to facilitate better diabetes outcomes. Healthcare workers should involve family and peers in diabetes education, care planning, and routine adherence, promoting motivation, emotional well-being, and accountability through peer support groups and community-based programs.

The synthesis of qualitative evidence also identified six overarching themes that facilitate effective diabetes self-care among patients in Ethiopia: social and family support, health system enablers, individual capacities, religious and cultural strengths, and systemic readiness [21–23,25,31–37].

Consistently across studies, family and social support were identified as key facilitators of diabetes self-care [22,23,25,34]. Patients reported that close family members, particularly children and spouses, played an instrumental role in reminding them to take medications, assisting with dietary choices, and providing emotional reassurance [22,25]. Healthcare providers also noted the positive impact of involving caregivers in educational initiatives, with better outcomes observed when both the patient and their caregiver were educated [23]. This finding aligns with studies done in Kenya [40], India [41], and Nigeria [42,43], sharing similarities to the Ethiopian experience. Thus, Healthcare systems should involve families and caregivers in treatment and education programs to improve health outcomes, strengthen self-management efforts, and recognize the role of social networks in long-term behavior change.

The health system plays a crucial role in promoting diabetes self-care through targeted education and consistent follow-ups, with individualized counseling and continuous care enhancing patient compliance with diabetes management routines [21,25,37]. Similar findings have been reported in Ethiopia [64], Nigeria [43], India [41], the UK [65,66], southern California [67], the Philippines [68], and Australia [69].this due to health system's role in diabetes management is often underdeveloped due to resource constraints, inadequate staffing, and lack of training. Integrating diabetes care into primary health services and using digital tools for reminders and remote support can enhance care continuity and patient engagement.

Individual factors such as health knowledge, practical skills, and confidence were identified as significant enablers of diabetes self-care [22,32,34]. Patients with higher educational levels or prior training were more capable of following self-care practices [22,32]. This is consistent with studies in other contexts, such as in Pakistan [70] and India [41]. The confidence patients exhibited in managing their care was similarly noted in Ethiopia, where individuals reported that using self-monitoring devices, such as glucometers, facilitated better management [34]. This observation mirrors findings

from Egypt [57] and the USA [58]. Healthcare providers should enhance diabetes self-care health literacy, develop self-management skills, and boost patient confidence through continuous education, accessible materials, and ongoing reinforcement through community health workers or digital platforms.

Interestingly, some studies found that religious and cultural practices, often perceived as barriers, could also serve as facilitators when aligned with self-care behaviors [31,36]. In the Ethiopian context, a patient described how religious hygiene routines, such as frequent foot washing for prayer, naturally complemented clinical foot care recommendations [36]. Similarly, another participant linked traditional views of stress and spiritual well-being with modern psychological strategies, noting the importance of managing stress to maintain health [31]. These findings echo similar insights from studies done in Malaysia [71], China [72], and Canada [73]. A Pakistani study found that patients view Islamic practices like fasting and prayer as spiritual obligations, enhancing diet and medication adherence [74]. In Indonesia, religious beliefs and communal values boost patient motivation and emotional support in managing diabetes [75]. These cross-cultural parallels emphasize the need for healthcare systems to integrate patients' religious and cultural frameworks into diabetes education and care models, recognizing these practices as enablers to improve engagement, adherence, and overall diabetes outcomes in diverse sociocultural settings.

Signs of systemic readiness emerged as crucial facilitators of diabetes self-care, particularly within tertiary healthcare settings in Ethiopia, where specialized services and policy-level support were evident [21,23]. Participants expressed optimism about the evolving healthcare environment, with stakeholders showing interest in developing tailored [21], patient-centered care models and improvements in health system infrastructure, such as medical tools and trained professionals, which positively influenced their ability to manage their condition [23]. These findings are consistent with research conducted in countries like India [76,77], South Africa [78], Ghana [79], and Australia [80]. These global parallels highlight the importance of health system readiness, including infrastructure, trained personnel, and patient-centered policy development as a foundational enabler for effective diabetes self-care. High-income countries like the UK, Australia, and Canada [49,81] have well-established systems for diabetes self-care, including patient-centered care, robust infrastructure, and trained multidisciplinary teams. However, Ethiopia faces challenges in maintaining these enablers due to variations in health system financing, policy implementation, and resource allocation.

## Strengths and limitations

This systematic review has several notable strengths. It is the first of its kind to comprehensively synthesize qualitative evidence on barriers and enablers to diabetes self-care in Ethiopia, offering rich, context-specific insights drawn from diverse regions and populations. The use of rigorous methodologies, including adherence to PRISMA 2020 and PRISMA-QS guidelines, application of the Joanna Briggs Institute (JBI) methodology for qualitative systematic reviews, and the GRADE-CERQual approach for assessing confidence in review findings, enhances the credibility and trustworthiness of the synthesis. Furthermore, the inclusion of both patient and healthcare provider perspectives provides a multidimensional understanding of the challenges and opportunities in diabetes self-management. The review has limitations, including geographical underrepresentation of rural communities and a lack of detailed reporting of ethical procedures. Despite these, the findings offer valuable implications for improving diabetes care in Ethiopia and similar low-resource settings, affecting practice, policy, and future research. At the review level, limitations include the inclusion of only English-language publications and potential publication bias, which may have excluded relevant unpublished or non-English studies. At the evidence level, many included studies underrepresented rural populations and provided limited detail on ethical procedures and researcher reflexivity, which may affect transferability.

## Conclusion and recommendations

This systematic review provides an in-depth understanding of the multifaceted barriers and enablers of diabetes self-care in Ethiopia. The findings reveal that self-care practices are significantly influenced by a complex interplay of individual,

cultural, social, and systemic factors. Key barriers include limited access to medical supplies, poor health literacy, psychological distress, socioeconomic hardship, and deeply rooted cultural beliefs. Conversely, strong family support, targeted health education, motivated patients, culturally aligned practices, and signs of systemic readiness emerged as critical enablers. These insights underscore the need for a holistic, patient-centered approach to diabetes management in Ethiopia. The study suggests Ethiopia should strengthen its health system by ensuring consistent availability of medications, diagnostic tools, and trained personnel, integrating culturally sensitive diabetes education into routine care, and developing national strategies to improve access to self-care resources, especially in underserved rural areas.

## Supporting information

**S1 File. List of articles excluded in the final studies exploring barriers and enablers to diabetes self-care in Ethiopia, 2025.**
(DOCX)

**S2 File. PRISMA 2020 checklist demonstrating adherence to the preferred reporting items for systematic reviews and meta-analyses (PRISMA) 2020 guidelines and the PRISMA extension for qualitative evidence synthesis (PRISMA-QS) throughout all stages of the review process on exploring barriers and enablers to diabetes self-care in Ethiopia, 2025.**
(DOCX)

## Author contributions

**Conceptualization:** Sadik Abdulwehab.

**Data curation:** Sadik Abdulwehab, Frezer Kedir.

**Formal analysis:** Sadik Abdulwehab, Frezer Kedir.

**Investigation:** Sadik Abdulwehab, Frezer Kedir.

**Methodology:** Sadik Abdulwehab, Frezer Kedir.

**Project administration:** Sadik Abdulwehab.

**Software:** Sadik Abdulwehab, Frezer Kedir.

**Supervision:** Sadik Abdulwehab.

**Validation:** Sadik Abdulwehab, Frezer Kedir.

**Visualization:** Sadik Abdulwehab.

**Writing – original draft:** Sadik Abdulwehab, Frezer Kedir.

**Writing – review & editing:** Sadik Abdulwehab, Frezer Kedir.

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
