## [Decision Letter · Decision Letter 0]

19 Dec 2025

Thank you for submitting your manuscript to PLOS ONE. After careful consideration, we feel that it has merit but does not fully meet PLOS ONE’s publication criteria as it currently stands. Therefore, we invite you to submit a revised version of the manuscript that addresses the points raised during the review process.

We look forward to receiving your revised manuscript.

Kind regards,

Kahsu Gebrekidan, Ph.D.

Academic Editor

PLOS One

Journal Requirements:

3. Please include a separate caption for figure in your manuscript.

Reviewers' comments:

Reviewer's Responses to Questions

**Comments to the Author**

1. Is the manuscript technically sound, and do the data support the conclusions?

Reviewer #1: Partly

Reviewer #2: Yes

2. Has the statistical analysis been performed appropriately and rigorously?

Reviewer #1: N/A

Reviewer #2: N/A

3. Have the authors made all data underlying the findings in their manuscript fully available?

Reviewer #1: Yes

Reviewer #2: Yes

4. Is the manuscript presented in an intelligible fashion and written in standard English?

Reviewer #1: No

Reviewer #2: Yes

Reviewer #1: Summary of Key Strengths

Clear alignment with PRISMA, PRISMA-QS, JBI standards.

Strong methodological transparency and reflexivity.

Comprehensive search strategy and explicit inclusion criteria.

Appropriate use of thematic synthesis and CERQual.

Rigorous dual-review processes.

Priority Areas for Improvement

Reduce minor redundancies to increase conciseness.

Clarify reviewer roles in coding and appraisal (independent vs. collaborative).

Consider reorganizing the Search Strategy and SPIDER criteria for smoother readability.

Move detailed timelines to an appendix to streamline the Methods narrative.

Reviewer #2: Exploring Barriers and Enablers to Diabetes Self-Care Practices in Ethiopia, A

Qualitative Systematic Review

This qualitative systematic review sought to synthesise qualitative evidence on the barriers and facilitators of diabetes self-care practices in Ethiopia.

The issues identified for review by the authors have been provided below.

Abstract

• In line with PRISMA for abstracts, specify the review’s inclusion and exclusion criteria further.

• Specify the type of studies in the statement: “The database was searched for every article published on of diabetes self-care practices”. Revise the placement of “of” in the sentence.

• Provide a succinct summary of the relevant characteristics of the included studies.

Introduction

• Revise the expression “diabetic patients” which could be deemed dehumanising/stigmatising (https://www.bmj.com/content/390/bmj.r981). Consider using “person with diabetes” or “people with diabetes”.

• Introduce the reader to what self-care is per this study.

Methods

• Aim: Align the aim in the methods with the aim in the abstract and introduction.

• Search: Specify whether each reviewer reviewed 100% of all title + abstracts and full texts.

• Data Extraction: Note how disagreements regarding the data extracted were resolved between the reviewers.

• Describe any assumptions made about any missing or unclear information. Note whether authors were contacted regarding missing information.

Results

• Indicate the total number of records identified in the search, titles and abstracts before the full texts screened.

• Provide in the appendix, the list of studies excluded from the review and the rationale for exclusion.

• Indicate the year ranges of the included studies.

• Align the names of the themes in the Findings with those in Table Four.

Eg. “Health system limitations” in stated the results but written as “Health System Factors”,

“Individual-level challenges” in Results vs “Individual Factors” in Table 4., “Socioeconomic constraints” in results vs “Socioeconomic Barriers”.

• Synthesis of Facilitators to Diabetes Selfcare: Reconcile the number of themes indicated “six overarching themes” vs the number of themes in Table 5, which are five.

Discussion

• Revise the statement “this finding similar with other study done in Ethiopia…………”

• Discuss other literature that was contrary to the review findings. Provide a possible reason for the difference in findings.

• Specify which “the Joanna Briggs Institute framework” is referred to in the strengths and limitations.

• The limitation statement “The review has limitations, including geographical underrepresentation of rural communities and lack of detailed ethical procedures reporting” is inadequate. Separate and clarify the limitations of the review from those of the evidence.

.

Reviewer #1: No

Reviewer #2: No

---

## [Author Response · Author response to Decision Letter 1]

23 Dec 2025

Response for reviewers

Response for reviewer 1

We sincerely thank Reviewer #1 for the constructive comments and valuable suggestions. We carefully considered each point and have made revisions to improve the clarity, conciseness, transparency, and readability of the manuscript. Detailed responses and corresponding manuscript changes are provided below.

Reviewer #1:

Comment 1: Reduce minor redundancies to increase conciseness

Reviewer comment:

Reduce minor redundancies to increase conciseness.

Response: We thank the reviewer for this helpful suggestion. We carefully reviewed the manuscript to identify and remove minor redundancies that affected conciseness and readability.

Manuscript changes: Redundant descriptions were reduced across the Abstract, Introduction, Results, and Discussion sections. In particular, overlapping explanations of diabetes self-care importance and repeated descriptions of key themes (e.g., health system limitations and social support) were streamlined. The Discussion section was revised to focus more on interpretation and implications rather than restating Results. These revisions improved clarity while preserving analytical depth.

#COMMENT 2: Clarify reviewer roles in coding and appraisal (independent vs. collaborative)

Reviewer comment:

Clarify reviewer roles in coding and appraisal (independent vs. collaborative).

Response: We appreciate the reviewer’s emphasis on methodological clarity. We have revised the Methods section to explicitly describe the independent and collaborative roles of reviewers throughout screening, data extraction, coding, synthesis, and quality appraisal.

Manuscript changes: Explicit statements have been added to the Search and Screening Process, Data Extraction, Data Synthesis, and Quality Appraisal sections clarifying that reviewers worked independently at initial stages, followed by consensus-based discussions to resolve discrepancies. These revisions strengthen transparency and rigor in line with PRISMA-QS and JBI guidance.

We revise each section as below

Search and Screening Process-Two reviewers (SA and FK) independently screened titles and abstracts. Full-text screening was also conducted independently. Discrepancies were resolved through discussion and consensus.

Data Extraction-Data extraction was conducted independently by two reviewers using a standardized form. Extracted data were compared, and any inconsistencies were resolved through discussion.

Data Synthesis-Initial line-by-line coding was performed independently by both reviewers. Codes were then compared and collaboratively refined into descriptive and analytical themes.

Quality Appraisal-Two reviewers independently appraised all included studies using the CASP Qualitative Checklist. Disagreements were resolved through discussion, with consensus reached in all cases.

#COMMENT 3: Reorganize Search Strategy and SPIDER criteria for readability

Reviewer comment:

Consider reorganizing the Search Strategy and SPIDER criteria for smoother readability.

Response: We thank the reviewer for this constructive suggestion. To enhance readability and logical flow, we reorganized the Search Strategy and Inclusion Criteria sections.

Manuscript changes:

The Search Strategy section was streamlined to focus on databases, search terms, and search procedures. The SPIDER framework was clearly presented within the Inclusion and Exclusion Criteria section, with each component explicitly described. This reorganization improves clarity while maintaining methodological completeness.

#COMMENT 4: Move detailed timelines to an appendix

Reviewer comment:

Move detailed timelines to an appendix to streamline the Methods narrative.

Response: We agree with the reviewer that excessive procedural detail can interrupt narrative flow.

Manuscript changes: Detailed timelines related to data extraction, synthesis, and report preparation were removed from the main Methods section and relocated to an appendix (Supplementary Material). The Methods narrative was streamlined accordingly.

Overall, these revisions were made to improve clarity, transparency, and readability while maintaining alignment with PRISMA, PRISMA-QS, and JBI standards. We believe that the revisions made in response to Reviewer #1 have strengthened the manuscript by enhancing methodological clarity, streamlining content, and improving the presentation of results and discussion. We appreciate the reviewer’s guidance, which has substantially contributed to the overall quality and readability of our work.

Response to Reviewer #2

We sincerely thank Reviewer #2 for the thoughtful and detailed feedback on our manuscript. We have carefully considered each comment and made revisions to improve clarity, transparency, methodological rigor, and alignment with reporting guidelines. Detailed responses and corresponding manuscript changes are provided below.

Reviewer #2: Exploring Barriers and Enablers to Diabetes Self-Care Practices in Ethiopia, A

Qualitative Systematic Review. This qualitative systematic review sought to synthesise qualitative evidence on the barriers and facilitators of diabetes self-care practices in Ethiopia.

The issues identified for review by the authors have been provided below.

#1. Reviewer comment: In line with PRISMA for abstracts, specify the review’s inclusion and exclusion criteria further; specify the type of studies searched; and provide a succinct summary of the relevant characteristics of the included studies.

Response:

Thank you for this important suggestion. The Abstract has been revised in accordance with PRISMA for Abstracts to improve clarity, transparency, and completeness.

Manuscript changes: The Methods subsection of the Abstract now explicitly states the inclusion and exclusion criteria, clearly specifying that primary qualitative and mixed-methods studies with extractable qualitative findings conducted in Ethiopia were included, while quantitative studies, reviews, and studies conducted outside Ethiopia were excluded. The description of the search strategy was revised to clearly indicate the type of studies sought and to correct wording and sentence structure. In addition, the Results subsection now includes a concise summary of key characteristics of the included studies, including study design, publication period, settings, and participant groups.

Introduction

#2. Response to Reviewer – Introduction (Language and Conceptual Clarity)

Reviewer comment:

Revise the expression “diabetic patients,” which could be deemed dehumanising/stigmatising, and introduce the reader to what self-care is per this study.

Response:

We thank the reviewer for highlighting this important issue. In response, we revised the manuscript to consistently adopt person-first, non-stigmatizing language in line with BMJ recommendations. Expressions such as “diabetic patients” were replaced with “people living with diabetes” or “people with diabetes” throughout the Introduction and manuscript. In addition, we introduced a clear, study-specific definition of diabetes self-care in the Introduction to orient readers to how the concept is understood and applied in this review.

Manuscript changes: Person-first language was implemented across relevant sections, including the description of study populations. A concise definition of diabetes self-care was added following the paragraph describing self-care behaviors to enhance conceptual clarity.

#3. Response to Reviewer #1 – Methods Section

Reviewer comment:

Align the aim in the Methods with the aim in the Abstract and Introduction.

Response:

The aim of the study has been revised to ensure full consistency across the Abstract, Introduction, and Methods sections.

Manuscript changes:

The Aim subsection in the Methods was reworded to explicitly state that the review seeks to generate a contextualized and comprehensive understanding of barriers and facilitators influencing diabetes self-care practices in Ethiopia through synthesis of qualitative and mixed-methods evidence.

Reviewer comment: Search process

Specify whether each reviewer reviewed 100% of all titles/abstracts and full texts.

Response:

We have clarified the screening responsibilities of the reviewers to enhance methodological transparency.

Manuscript changes:

The Search and Screening Process subsection now explicitly states that two reviewers independently screened 100% of all titles and abstracts, as well as all full-text articles, with disagreements resolved through discussion and consensus.

Reviewer comment: Data extraction

Note how disagreements regarding the data extracted were resolved between the reviewers.

Response:

This detail has now been clearly described in the Methods section.

Manuscript changes:

The Data Extraction subsection was revised to specify that data extraction was conducted independently by two reviewers, after which extracted data were compared and any discrepancies were resolved through discussion and consensus.

Reviewer comment: Missing or unclear information

Describe any assumptions made about missing or unclear information and note whether authors were contacted.

Response:

We appreciate this suggestion and have added clarification to improve transparency.

Manuscript changes: A statement was added indicating that when information was missing or unclear, interpretation was limited to the data explicitly reported in the studies, no assumptions were made beyond the available information, and study authors were not contacted as all included studies provided sufficient qualitative data for synthesis.

Results

Reviewer Comment 1-Indicate the total number of records identified in the search, titles and abstracts before the full texts screened.

Response:

We have revised the Study Selection subsection of the Results to clearly report the number of records identified through database searching (n = 62) and other sources (n = 2), the total records screened at the title and abstract level (n = 64), the number of full-text articles assessed for eligibility (n = 26), and the final number of studies included in the synthesis (n = 11). These details are now explicitly stated and aligned with Figure 1 (PRISMA flow diagram).

Reviewer Comment 2-Provide in the appendix the list of studies excluded from the review and the rationale for exclusion.

Response: We have added a statement in the Results section indicating that studies excluded after full-text review (n = 15) and their reasons for exclusion are provided in Supplementary Table 1. The appendix now contains a detailed list of excluded studies with clear justifications for exclusion.

Reviewer Comment 3-Indicate the year ranges of the included studies.

Response: The Results section has been updated to include the publication year range of the included studies. This information is now reported in the Study Selection subsection to improve transparency regarding the temporal scope of the evidence.

Reviewer Comment 4-Align the names of the themes in the Findings with those in Table Four (e.g., “Health system limitations” vs “Health System Factors”).

Response: We have carefully reviewed and standardized all theme labels in the Results section to ensure consistency with the corresponding tables. For example, “Health system limitations” has been revised to “Health System Factors,” “Individual-level challenges” to “Individual level Factors,” and “Socioeconomic constraints” to “Socioeconomic Barriers.” This alignment has been applied throughout the text and tables.

Reviewer Comment 5-Synthesis of Facilitators to Diabetes Selfcare: Reconcile the number of themes indicated “six overarching themes” vs the number of themes in Table 5, which are five.

Response: We have corrected this inconsistency by revising the text to report five overarching facilitator themes, consistent with Table 4. The Results section now accurately reflects the number and naming of facilitator themes presented in the table.

Discussion

Reviewer Comment 1

Revise the statement “this finding similar with other study done in Ethiopia…”

Response: We revised all such statements to improve academic tone and clarity by using standardized scholarly language when comparing findings with previous studies.

Reviewer Comment 2

Discuss literature that is contrary to the review findings and explain possible reasons.

Response: We added discussion of contrasting findings from other settings and provided contextual explanations for differences, including variations in health system capacity, socioeconomic conditions, cultural contexts, and access to diabetes care.

Reviewer Comment 3

Specify which Joanna Briggs Institute framework was used.

Response: We clarified that the review followed the Joanna Briggs Institute (JBI) methodology for qualitative systematic reviews, including the JBI Critical Appraisal Checklist for Qualitative Research.

Reviewer Comment 4

Clarify and separate limitations of the review from limitations of the evidence.

Response: We revised the Strengths and Limitations section to clearly distinguish between limitations of the review process and limitations inherent in the included studies.

We believe that the revisions made in response to Reviewer #2 have strengthened the manuscript by enhancing clarity, ensuring consistency across sections, and improving methodological transparency. We appreciate the reviewer’s guidance, which has contributed substantially to the overall quality and readability of our work.

---

## [Decision Letter · Decision Letter 1]

26 Mar 2026

Exploring Barriers and Enablers to Diabetes Self-Care Practice in Ethiopia, 2025: A Qualitative Systematic Review

PONE-D-25-49603R1

Dear Dr. Sadik,

We’re pleased to inform you that your manuscript has been judged scientifically suitable for publication and will be formally accepted for publication once it meets all outstanding technical requirements.

Kind regards,

Kahsu Gebrekidan, Ph.D.

Academic Editor

PLOS One

Additional Editor Comments (optional):

Reviewers' comments:

Reviewer's Responses to Questions

**Comments to the Author**

Reviewer #2: All comments have been addressed

Reviewer #3: All comments have been addressed

2. Is the manuscript technically sound, and do the data support the conclusions?

Reviewer #2: Yes

Reviewer #3: Yes

3. Has the statistical analysis been performed appropriately and rigorously?

Reviewer #2: N/A

Reviewer #3: Yes

4. Have the authors made all data underlying the findings in their manuscript fully available?

Reviewer #2: Yes

Reviewer #3: Yes

5. Is the manuscript presented in an intelligible fashion and written in standard English?

Reviewer #2: Yes

Reviewer #3: Yes

Reviewer #2: Thank you for addressing the relevant issues raised which has improved the clarity and rigour of the review.

Reviewer #3: The authors have addressed and responded to all the concerns raised. Overall, the revision demonstrates a constructive effort to address the concerns raised during the initial review. The manuscript has improved in clarity, structure, and scientific rigor. Below is an evaluation of how effectively each major issue was handled:

Major Comments

Comment 1: Reduce minor redundancies to increase conciseness- The authors have adequately addressed this concern and the revisioin is satisifactory.

COMMENT 2: Clarify reviewer roles in coding and appraisal (independent vs.

collaborative). The response clearly stated the roles.

COMMENT 3: Reorganize Search Strategy and SPIDER criteria for readability. This has been addressed in the responses given and it is satisfactory

COMMENT 4: Move detailed timelines to an appendix.

The authors have addressed this concern accordingly.

.

Reviewer #2: No

Reviewer #3: No

---

## [Editor Report · Acceptance letter]

PONE-D-25-49603R1

PLOS One

Dear Dr. Abdulwehab,

I'm pleased to inform you that your manuscript has been deemed suitable for publication in PLOS One. Congratulations! Your manuscript is now being handed over to our production team.

Kind regards,

on behalf of

Dr. Kahsu Gebrekidan

Academic Editor

PLOS One